# Work Motivation: A Wall That Not Even the COVID-19 Pandemic Could Knock Down: Research Article

**DOI:** 10.3390/healthcare12181857

**Published:** 2024-09-15

**Authors:** Patrik M. Bogdán, Miklós Zrínyi, Ildikó Madarász, Lívia Tóth, Annamária Pakai

**Affiliations:** 1Doctoral School of Health Sciences, Faculty of Health Sciences, University of Pécs, 7621 Pécs, Hungary; ildiko.madarasz@etk.pte.hu (I.M.); toth.livia@pte.hu (L.T.); 2Institute of Emergency Care, Pedagogy of Health and Nursing Sciences, Faculty of Health Sciences, University of Pécs, 7621 Pécs, Hungary; miklos.zrinyi@etk.pte.hu (M.Z.); annamaria.pakai@etk.pte.hu (A.P.)

**Keywords:** COVID, nurse, work motivation, turnover

## Abstract

The emergence of the coronavirus pandemic in 2020 posed a new challenge, imposing extraordinary physical and psychological burdens on healthcare workers, clearly exacerbating and intensifying career abandonment. **Objectives:** The aim of our study was to explore the motivating factors among nurses serving during the coronavirus pandemic that they considered important in their profession despite the mental and physical stress brought about by the pandemic. **Methods:** A descriptive, cross-sectional study was conducted at the University of Pécs-Clinical Center-Regional Coronavirus Care Center between September 2022 and December 2022. We used non-random, expert, purposive sampling, recruiting healthcare workers who had spent at least 3 months working in a COVID ward (*n* = 196). Data collection was conducted by using an online, anonymous questionnaire, which included sociodemographic questions, the “Motivation at Work Scale”, and a self-edited six-item questionnaire. **Results:** Regarding the 5-year probability of remaining in the healthcare field, nine participants (4.5%) will definitely leave the healthcare sector, twenty-seven participants (13.7%) are undecided, and seventy-eight participants (39.7%) will definitely stay in the healthcare field over the next 5 years. There is a positive, weak, but significant correlation between intrinsic motivation and the probability of leaving the profession within 5 years (*r* = 0.281; *p* < 0.05). We identified a significant, negative, and weak correlation between the number of revisited waves of the coronavirus and the fear of redeployment to the COVID ward (*r* = −0.273; *p* < 0.05). **Conclusions:** Despite the challenges posed by the coronavirus pandemic, only a small percentage of nurses consider leaving the healthcare profession. Joy and enjoyment in their work were dominant factors even during the pandemic.

## 1. Introduction

### 1.1. Turnover

Among the helping professions, nursing is an area significantly affected by career abandonment [1,2]. This phenomenon is no longer exclusive to the elderly population, as career abandonment is also observed among recently graduated young professionals [3]. It may not be an exaggeration to state that this issue has become a pressing, daily problem worldwide [4]. In Europe, there are currently 7.3 million registered healthcare workers and midwives, but their numbers are still insufficient considering the demands placed on the healthcare system. Considering human resource data, it can be said that on average, there are 8.4 nurses per 1000 inhabitants in European countries. In terms of rankings, Germany ranks first with 13.9 nurses per 1000 inhabitants, while in Hungary, this number is only 6.5 nurses per 1000 inhabitants [5]. It cannot be emphasized enough that there is an increasing need for well-trained, young workforce replenishment, as the future continuously imposes growing demands [6]. Currently, there is a global shortage of 5.9 million nurses in the system, with the most severe shortages affecting East Asian and African countries. In the year 2020, the nurse turnover rate in the United States was 18.7% [7]. In the context of European countries, Germany is facing a shortage of nearly 500,000 nurses in the system. Additionally, the average duration of stay in general nursing fields is only 6 years, while in elderly care, it is 10 years [8] In a study conducted in France, the turnover of intensive care nurses was the most significant. The annual turnover rate for specialized nurses was 24%, while it was 13% among auxiliary nurses [9]. According to the results of a 2023 study in Switzerland, 28% of nurses working in elderly care changed careers within 5–6 years after obtaining their qualifications, while the same figure for hospital nurses was 17% [10]. The turnover rate was 23% in Israel and 12.4% in South Korea [11]. From the data mentioned above, it is clear that retaining nurses and recruiting new staff has become one of the most critical factors for the healthcare system [12].

### 1.2. Motivation

The issue of career abandonment is significantly influenced by motivation, particularly work motivation [13,14]. The basis of this is the theory of self-determination, which has been a highly researched area over the past 40 years and its theoretical foundations are commonly applied in organizational psychology [15,16,17]. The self-determination theory (SDT) is a psychological framework that emphasizes the role of intrinsic motivation and the fulfilment of basic psychological needs in fostering work motivation. Developed by Edward Deci and Richard Ryan, SDT highlights the significance of three fundamental psychological needs: autonomy, competence, and relatedness.

According to the theory of self-determination in relation to work motivation, the greater the amount of work activities that satisfy innate psychological needs, the greater the employee’s internal motivation for working without external pressure. Work motivation—whose theoretical background was developed by Edward Deci and Richard Ryan in the mid-1980s—has been shown to contribute to work intention and workplace satisfaction [18]. Higher levels of motivation have a positive impact on career retention [19]. By career retention, we mean that the employee remains in their current position and has no intention of seeking employment in other fields [20]. Numerous studies have investigated the components of workplace motivation, and as a result, the factors that are fundamentally important are now well-known. The significance of meaningful work, a friendly work environment, and opportunities for learning and development during daily tasks are crucial for motivation [21]. It is extremely important to mention as a motivational factor that employees should be familiarized with the organization’s operations and culture, as well as how they fit into this picture. Involving staff in decision making is essential for them to feel like valuable team members. Rewarding hard work and dedication is crucial; however, care must be taken to ensure that work does not come at the expense of physical and mental well-being. A proper work–life balance is the key to achieving this.

In a study conducted by Zeng et al. in 2022 among nurses, the inclination towards healthcare work varied solely based on the strength of internal motivation, while external motivation had no effect at all [22]. In a study by Breed et al. in 2020 among nurse leaders, they examined which factors motivate them during the daily challenges of their work. According to their findings, nurse leaders were committed to their work because they considered it important. They viewed their work environment and tasks as exciting and full of challenges. These values also fall under the realm of internal motivation [23]. According to the findings of Deressa and Zeru’s 2019 study, the most important motivational factors for nurses included continuous managerial encouragement and recognition, as well as financial incentives. Additionally, they considered a good work environment, strong team spirit, patient satisfaction, and a love for nursing to be important as well [24].

### 1.3. Coronavirus Pandemic

The emergence of the coronavirus pandemic (COVID-19) in 2020 posed a whole new challenge, imposing extraordinary physical and psychological burdens on healthcare workers, clearly exacerbating and intensifying career abandonment [25,26]. Factors that increased intention to abandon the career during the pandemic period are known, such as continuous overtime work, coping with death, and increased workload [27,28,29]. Among nurses, persistent mental and physical stress has been shown to play a role as a risk factor in the development of conditions such as post-traumatic stress, depression, anxiety, or even alcoholism [30]. Some research results showed a 40% rate for those considering leaving their job, while 25% expressed a desire to abandon their career [31]. However, it is also worth paying attention to those workers who, despite the burdens caused by the pandemic, have not given up on their calling. The novelty of our study lies in focusing on the motivating factors that were present in the healthcare professionals’ value system related to their vocation even during the toughest period of the pandemic, which positively influenced the intention to abandon the career. There has not yet been an article exploring these factors. Our research aims to fill this gap in the literature.

### 1.4. Aim of Our Study

The aim of our study was to predict the role of motivating factors among nurses serving during the coronavirus pandemic that they considered important in their profession despite the mental and physical stress brought about by the pandemic. Another goal of our study was to examine to what extent certain motivating factors acted as protective factors against career abandonment.

## 2. Materials and Methods

We conducted a descriptive, cross-sectional study for our research. The University of Pécs-Clinical Center-Regional Coronavirus Care Center served as the location for the study. Data collection took place between September 2022 and December 2022.

### 2.1. Ethical Consideration

Prior to initiating the research, we obtained ethical approval from the regional research ethics committee. (Ethical approval number: 9390-PTE 2022)

### 2.2. Sample Recruitment

We used non-random, expert, purposive sampling. We included male and female nurses who had spent at least 3 months working in a COVID ward. All participants were employees of the University of Pécs-Clinical Center, who were delegated to the regional coronavirus treatment center during the COVID-19 pandemic. Participants were recruited and the questionnaires were distributed through the university’s internal mailing system. Participation in the study was voluntary. The instructions for completing the questionnaire included the following statement: “By filling out and submitting the questionnaire, you consent to participate in the study.”.

### 2.3. Sample Size Calculation

A priori sample size calculation (GPower 3.1.9.7) assuming a low effect size (0.05), a significance level of 5% (0.05), a power of 20% (0.8), and with eleven predictors yielded a total required sample size of 126 subjects for the regression model (GPower, 2024). The final sample size of 186 subjects was sufficient to meet the a priori sample size criterion [32].

### 2.4. Data Collection

We conducted our data collection using an online questionnaire, which contained a total of 34 questions, and completing it took approximately 5 to 10 min. For better clarity, we recommended using a computer or tablet for filling it out. In addition to the sociodemographic data, we utilized the “Motivation at Work Scale” developed by Marylène Gagné and colleagues [33]. This questionnaire consists of 12 items that revolve around the four main categories of motivation: intrinsic motivation, identified regulation, introjected regulation, and external regulation. These categories were developed based on the multidimensional conceptualization theory outlined in the self-determination theory. Participants in the study were asked to indicate on a scale of 1 to 10 how true each statement in the questionnaire was for them. For example, “I do this work for the money” would be rated on a scale from 1 (not true at all) to 10 (completely true). The internal consistency of the subscales of the questionnaire is highest for intrinsic motivation, with a value of 0.89. The identified motivation has a value of 0.83, the introjected motivation is 0.75, while extrinsic motivation represents a Cronbach’s alpha value of 0.69. The Hungarian adaptation of the measurement tool was carried out according to professional standards. Three independent individuals translated the questionnaire from English into the target language. Then, during a personal discussion, we created a common version based on the three translations, which was subsequently backtranslated into the source language by a bilingual translator. We found no significant differences between the two original English versions, and thus we considered the adaptation to be complete. We conducted a pilot test with 20 participants using the Hungarian version of the questionnaire to determine whether the questions were uniformly interpretable and answerable for everyone. Additionally, we assessed the internal consistency of the adaptation. We obtained the following Cronbach’s alpha values: intrinsic motivation, 0.89; identified motivation, 0.868; introjected motivation, 0.804; and external regulation, 0.819.

We were also interested in understanding the level of concern among the participating nurses about being forced to work in a COVID ward again. Therefore, we created a custom questionnaire consisting of 6 questions. These questions included statements such as: “I feel bad when I think about being assigned to a COVID ward again”. Respondents were asked to rate their agreement with each statement on a Likert scale ranging from 0 to 4, depending on how true the statement was for them. The questionnaire possesses face validity. To validate the reliability of the questions, we sought the assistance of a psychologist who has worked in mental care with nurses affected during the COVID-19 pandemic. The professional reviewed our questions and deemed them appropriate for assessing the phenomenon under investigation. The measurement tool achieved a Cronbach’s alpha value of 0.918. The six items were summed and divided by the number of items to achieve a total scale score, which was used as a predictor in the multiple regression analysis.

Participants had to answer one of our custom questions, which asked them to rate on a scale of 0 to 10 the likelihood that they would continue working in the healthcare field in the next 5 years. A score of 1 indicated that they would definitely not continue working in healthcare within the next 5 years, while a score of 10 represented the highest likelihood.

### 2.5. Data Analysis

The Microsoft Office Excel 2016 and SPSS version 25 software package was used for data processing. Means, standard deviation, minimum and maximum score, and absolute and relative frequency were calculated in descriptive statistics. For statistical analysis, we used multiple linear regression and due to the non-normal distribution of our data, we applied non-parametric tests, such as Spearman’s correlation and the Mann–Whitney test.

### 2.6. Sample

A total of 196 nurses were involved. The descriptive statistics of the sample characteristics are presented in Table 1. The average age of the sample was 40.16 years (SD = 10.77). The average work experience was 17.86 years (SD = 11.97). The detailed sociodemographic data are presented in Table 1. The supplementary questions regarding the coronavirus pandemic are presented in Table 2.

Based on the results of the 5-year probability of remaining in the healthcare field, nine participants (4.5%) will definitely leave the healthcare sector, twenty-seven participants (13.7%) are undecided, and seventy-eight participants (39.7%) will definitely stay in the healthcare field over the next 5 years.

## 3. Results

### 3.1. Result of the Motivation at Work Scale

We provide the averages scores of “MAWS” scale questions in Figure 1.

There is a negative, negligible correlation between age and “I chose this job because it allows me to reach my life goals” (*r* = −0.178; *p* < 0.05). Similarly, there is a negative, negligible correlation between age and “Because this job fulfils my career plans” (*r* = −0.164; *p* < 0.05).

There is a negative, negligible correlation between the number of years worked in healthcare and “I chose this job because it allows me to reach my life goals” (*r* = −0.269; *p* < 0.05). Similarly, there is a negative, negligible correlation between the number of years worked in healthcare and “Because this job fulfils my career plans” (*r* = −0.249; *p* < 0.05).

The four subdimensions of the “MAWS” questionnaire and the average scores of the subscales are presented in Figure 2.


*Intrinsic motivation:*


The average scores of participants pursuing higher healthcare studies (8.3 ± 2.08 points) were significantly higher than those who did not (7.2 ± 2.28 points) (*p* = 0.001).

There is a positive, weak, but significant correlation between intrinsic motivation and the probability of leaving the profession within 5 years (*r* = 0.281; *p* < 0.05).

There was no correlation found between the subscale scores of intrinsic motivation on the “MAWS” questionnaire and the fear of being redelegated to a COVID ward (*p* > 0.05).

There was no correlation found between the average scores of intrinsic motivation and gender, age, educational background, income category, monthly hours worked in the COVID ward, number of night shifts, and number of COVID waves worked (*p* > 0.05).


*Identified Regulation:*


The average scores of participants pursuing higher healthcare studies (7.6 ± 2.1 points) were significantly higher than those who did not (6.3 ± 2.46 points) (*p* = 0.001).

There is a negative, weak but significant correlation between the scores of the identified regulation subscale and the number of years worked (*r* = −0.230; *p* = 0.001).

There is a significant, positive, weak correlation between the probability of leaving the profession within 5 years and the scores on the identified regulation subscale (*r* = 0.319; *p* < 0.05).

No correlation was found between the identified regulation subscale on the “MAWS” questionnaire and the fear of being redelegated to a COVID ward (*p* > 0.05).

There was also no correlation found between gender, age, educational background, income category, monthly hours worked in the COVID ward, number of night shifts, history of contracting the virus, experience of losing a close acquaintance, and number of COVID waves worked (*p* > 0.05).


*Introjected Regulation:*


There is a positive, weak, almost negligible correlation between the average scores of the subscale and the probability of leaving the profession within 5 years (*r* = 0.190; *p* = 0.015).

No correlation was found between gender, age, educational background, current studies, income category, years worked, monthly hours worked in the COVID ward, number of night shifts, or number of COVID waves worked (*p* > 0.05).

There was no correlation found between this subscale and the fear of being redelegated to a COVID ward (*p* > 0.05).


*External Regulation:*


The average scores of the subscale were significantly higher (5.19 ± 2.11 points) among those in the income category of 500,000 HUF or above, compared to those earning 200,000 HUF or below (2.1 ± 1.77 points) (*p* = 0.021).

There is a positive, weak, almost negligible correlation between the average scores of the external regulation subscale and the fear of being redelegated to a COVID ward (*r* = 0.169; *p* = 0.018).

No correlation was found between gender, age, educational background, current studies, income category, years worked, monthly hours worked in the COVID ward, number of night shifts, number of COVID waves worked, and the probability of leaving the profession within 5 years (*p* > 0.05).

The correlation data examining the relationship between certain questions from the Motivation at Work Scale questionnaire and the probability of staying in the healthcare field for 5 years can be found in Table 3.

### 3.2. Result of the “Fear of Reassignment to a COVID Ward” Questionnaire

The average scores measured in the questionnaire are presented in Figure 3.

In the reliability test conducted on the questionnaire, we obtained a Cronbach’s alpha value of 0.918. The average score for females (3.22 ± 1.24 points) was significantly higher than the average score for males (2.42 ± 1.2 points) (*p* < 0.05). The average score of those pursuing educational studies during the study (2.61 ± 1.25 points) was significantly lower when compared to the average score of those who were not pursuing educational studies (3.26 ± 1.23 points) (*p* = <0.05).

We identified a significant, negative, weak correlation between the number of revisited waves of the coronavirus and the fear of redeployment (*r* = −0.273; *p* < 0.05). A negative, weak, almost negligible, significant correlation can be observed between staying in the profession within 5 years and the fear of redeployment (*r* = −0.146; *p* < 0.05).

In terms of age, work experience, education level, income status, monthly working hours, monthly night hours, history of infection, staff size, and fear of redeployment, we could not identify any correlations (*p* > 0.05).

### 3.3. Multiple Linear Regression

Multiple linear regression was developed to predict the probability of remaining in healthcare for another 5 years. To refine the model, outliers with studentized residuals greater than ±2 were removed from further analysis. Dummy variables were created for each independent categorical variable.

In conclusion, a multiple linear regression model was constructed (Table 4) with the probability of remaining employed in the healthcare sector for a further five years as the dependent variable. Outliers exceeding the studentized residuals by a value of ± 2 were excluded from the final analysis. The final model was statistically significant (F = 5.69, *p* < 0.001), and the current set of independent variables explained 26.3% of the variance in the dependent variable. The three independent variables that reached significance, in the order of their relative contribution to the regression model (beta weights), were identified as motivation, length of working in healthcare, and personal age. A one-point increase in identified motivation was associated with a 0.427-point increase in the probability of remaining in the healthcare sector for an additional five years. For each additional year spent working in healthcare, the probability of remaining in the same position for the following five years increased by 0.09 points. However, age was inversely related to intention to remain in the healthcare sector over the following five years. For each additional year of age, the probability of remaining in the sector was reduced by 0.088 points.

## 4. Discussions

The aim of our study was to identify the motivating factors among nurses serving during the coronavirus pandemic, which they considered important in their profession despite the mental and physical stress caused by the pandemic. We aimed to examine to what extent certain motivating factors acted as protective factors against leaving the profession. We were also curious to see how anxious healthcare workers are about a possible redeployment to COVID care centers.

The issue of leaving the healthcare profession has been a long-standing and discussed topic [34]. Among the nurses we surveyed, seventy-eight out of one hundred ninety-six, that is, 39%, definitely do not plan to leave healthcare in the next 5 years, and only nine individuals, or 4.6%, thought they definitely do not want to stay in the profession. We believe that this number is very low, indicating that despite the coronavirus pandemic, only a small percentage of nurses are so negatively affected by the pandemic that they are considering leaving their profession. In the case of 27 individuals, or 13.7%, the response to leaving the profession was “either yes or no.” It would be worth observing and assessing the future factors that influence their decision making on this matter [35].

### 4.1. Intrinsic Motivation

The intrinsic motivation subscale of the Motivation at Work Scale questionnaire achieved the highest average score. We believe that nurses practice their profession primarily for the joy and pleasure it brings them, which served as a solid foundation even amidst the negative effects of the pandemic. Contrary results were obtained in a study conducted by Smokrović et al. in 2022 [36]. We found significantly higher scores on the intrinsic motivation subscale among those who pursued university studies, indicating that higher education and continuous self-improvement serve as demonstrable protective factors. This finding aligns with the research conducted by D’alleva A. et al. in 2023 [37]. We also determined that the more nurses possess intrinsic motivational factors, the lower the likelihood of leaving the profession within 5 years among them [38]. Based on our findings, we consider it is important for employers to ensure and support the fulfillment of intrinsic motivation and its deeper realization. In terms of educational qualifications and motivation, it is crucial for employers to provide continuous training opportunities for nurses. It is very important to support the professional development of younger generations through various mentoring programs, involving older and more experienced nurses. In addition to supporting the fulfillment of intrinsic motivation, establishing a balanced work schedule is also essential.

### 4.2. Identified Regulation

Identified motivation refers to the form of motivation where someone commits to a task or goal and understands and perceives its necessity, even if they have not made efforts in favor of the cause before. Our research findings reveal that the less time a nurse has been in the profession, the more likely they are to embrace these motivational factors. To maintain long-term motivation among nurses, it is important to understand the factors that contribute to decreasing motivation. Beyond the minimum necessary communication related to work, employers or direct supervisors must be aware of employees’ individual and personal issues or ambitions in relation to their profession. To achieve this, it is essential to provide opportunities for personal conversations or small group training sessions. The foundations of nursing as a helping profession must be established repeatedly, and nurses should be given the opportunity to reinforce within themselves why they chose this profession. It was also evident that nurses pursuing university studies exhibited a stronger identified motivational factor. This suggests that their participation in higher education can be interpreted as an understanding that education and self-improvement are in their best interest to stay current and credible in their profession [39].

### 4.3. Introjected Regulation

Translating our results for the period of the coronavirus pandemic, the stronger the introjected motivational forces that are present, the more likely the commitment towards a career in healthcare. We conclude that despite the challenges, nurses exhibited a desire to meet expectations and did not allow the negative effects to prevail over them. In other words, with this characteristic, they already possessed it before the pandemic and persevered through it. From a different perspective, we infer that the commitment to the profession was strengthened by the fact that nurses had to serve in an extremely challenging period, which enhanced their sense of “I will show what I am capable of” as a motivating force [40]. The driving force of introjected motivation is not internal intention and conviction but rather a kind of external pressure to achieve a goal. However, internal intention can be strengthened if we set a specific goal for the nurse (such as obtaining further qualifications, completing higher education, fulfilling educational responsibilities, or mentoring) and motivate them to achieve it with money or other rewards. It can have a positive effect on nurses if we motivate them with the successes of nurses who have had a great career.

### 4.4. External Regulation

Extrinsic motivation is determined by external factors. We engage in activities not for their own sake but in order to achieve a goal, reward, or success. We do things that we may not necessarily enjoy or find pleasure in, but we invest energy into the activity for the sake of the goal [41]. This phenomenon was highly observable in terms of income categories, as significantly higher scores were obtained on the subscale for higher income categories. Therefore, money as an external motivating force is dominantly present in the healthcare profession, understandably. It is important to emphasize this because our results clearly show that during the pandemic, money was indeed one of the most important motivating forces, as although nursing as a profession is based on internal values, livelihood, financial security, and financial development are also integral parts of motivations to stay in the field. Our results align with the research conducted by Negussie N. et al. in 2012 [42].

### 4.5. Factors of Remaining in the Profession within 5 Years

The strongest and most telling result of our study is that although we examined four dimensions of motivation, only the strength of identified motivation had a positive effect on the retention of nurses in the profession. In our study, identified motivational factors referred to the fulfilment of life goals, career-related plans, and personal values. In other words, if we want to strengthen nurses’ intention to remain in the profession, it is now clear that we need to focus on reinforcing these factors. Employers also play a significant role in achieving life goals, as we have emphasized earlier. Ambitious employees need to be provided with continuous development opportunities that align with nurses’ career plans. Development opportunities refer not only to the additional expertise that can be acquired through further education but also to enabling determined and skilled nurses to try out leadership roles. Personal values are, of course, unique to every individual, as they are shaped by both learned experiences and the interaction of numerous external and internal factors. Since they have a positive impact on retention in the healthcare profession, they are of great significance. Our suggestion is to identify and attract individuals to the healthcare system who have these dominant motivational components. The most important setting and target group for this are schools and young people who are about to make career choices. Although more professional experience positively influenced the intention to remain in the profession, higher age had a negative impact on it. In the context of the coronavirus pandemic, this phenomenon may be explained by the fact that nurses who had acquired greater professional knowledge over the years were less daunted by significant burdens and challenges, as they were well-prepared for patient care. This was crucial in the care of patients infected with the coronavirus, as the specifics of treating such patients were unknown to the medical field and remain unclear to this day. Our suggestion is that, in the event of a potential future pandemic, greater emphasis should be placed on selecting nurses with higher work experience. Moreover, if possible, the delegation of older age groups should be avoided, as our results clearly indicate that this could positively influence thoughts of leaving the profession.

### 4.6. Fear of Redeployment to a COVID Unit

We came to an interesting conclusion regarding the fear of redelegation, as it was least true for those we surveyed that they would not be able to go through another wave of the COVID pandemic; however, the statement that they feel bad at the thought of being redelegated to work on a COVID ward was most true for them. Based on this, we concluded that nurses were willing to be continuously present in COVID wards as the waves of the pandemic followed one another, but if the pandemic were to end, they would feel bad about having to put on the gloves again and continue their work in a COVID treatment center.

Female nurses have significantly higher fear of redelegation compared to men. This may be due to the fact that, according to the results of international research, women are exposed to higher levels of stress in their work compared to male nurses, so their anxiety about redelegation may also be higher [43].

It is an exciting finding that the fear of redelegation decreased with the number of worked COVID pandemics. It is unquestionable that the background to this is the experience gained and confidence in handling patients infected with the coronavirus. Initially, COVID treatment centers were surrounded by mystery and a kind of mystique, as media representatives were not allowed to enter these wards to minimize the risk of infection. Therefore, society could only learn about what a COVID ward was like from hearsay and shared news, which carried the possibility of spreading misinformation [44,45]. Our fear of the unknown is a fundamental human trait [46]. Based on this conclusion, we emphasize the importance of training and informing nurses on specific patient care and the characteristics of specialized treatment centers in the event of a future pandemic. It would have been beneficial to provide pre-visit opportunities for nurses designated for redelegation to COVID wards so that they would not arrive to their first shift filled with fear and doubts [47].

It is important to highlight in our finding that higher levels of anxiety about redelegation are associated with a higher likelihood of career abandonment. This finding draws attention to how crucial it is to delegate nurses to COVID wards in a rotating system, as this allows them time to rest, recover from emotional and mental overload, and regenerate. This could be a guarantee for reducing nurses’ abandonment of the healthcare profession.

## 5. Conclusions

Despite the challenges posed by the coronavirus pandemic, only a small percentage of nurses consider leaving the healthcare profession. Joy and enjoyment in their work were dominant factors even during the pandemic. Education and continuous improvement are essential for high-quality patient care and serve as protective factors against leaving the profession. Money, alongside personal values, plays a crucial role in motivating nurses to stay in their profession. The fear of redeployment to COVID wards is significant, especially among women, and it correlates with a higher likelihood of leaving the profession within 5 years. Delegating nurses with higher work experience is less risky, as it does not increase the intention to leave the profession. However, delegating older age groups is risky because it positively influences thoughts of leaving the nursing profession.

## 6. Limitations

The authors acknowledge the study limitations. It would have been beneficial to recruit a higher number of participants and include employees from small urban hospitals for interesting comparisons as their financial capacities and human resources differed from larger COVID centers during the pandemic. Since we applied non-random sampling, we caution readers about the immediate acceptance of our results.

## Figures and Tables

**Figure 1 healthcare-12-01857-f001:**
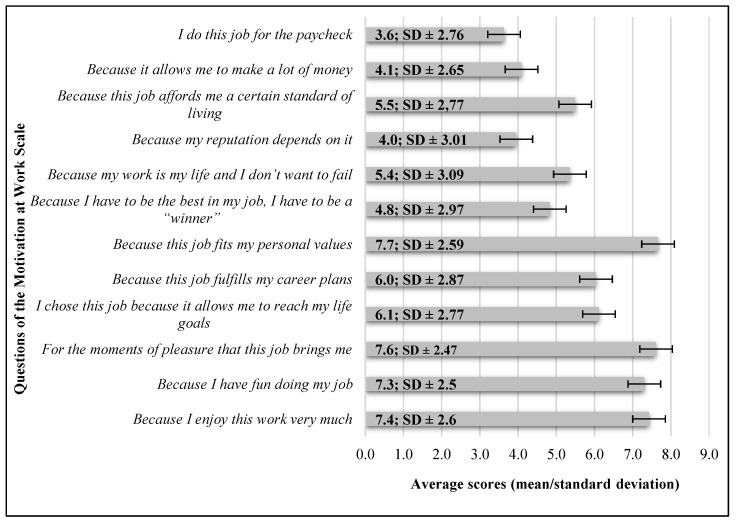
Average scores of the Motivation at Work Scale. (*n* = 196).

**Figure 2 healthcare-12-01857-f002:**
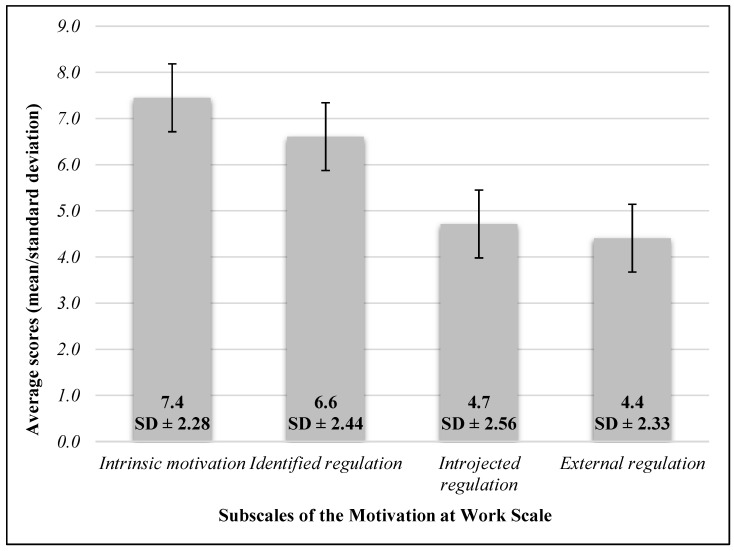
Subscales of the Motivation at Work Scale (*n* = 196).

**Figure 3 healthcare-12-01857-f003:**
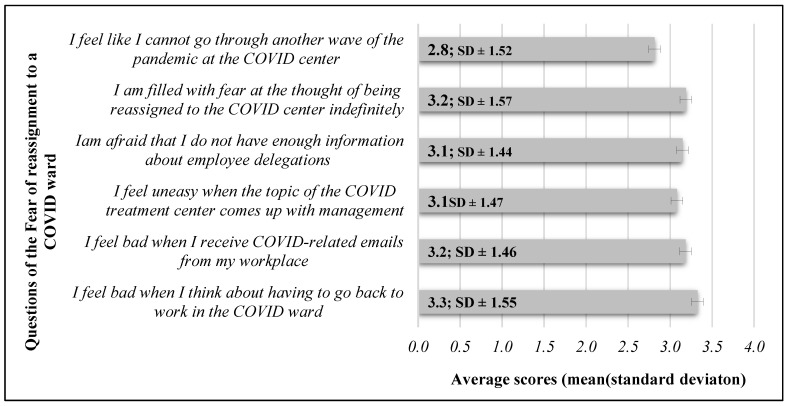
Results of the Fear of reassignment to a COVID ward questionnaire (*n* = 196).

**Table 1 healthcare-12-01857-t001:** Descriptive statistics of the sample characteristics (*n* = 196).

	Total Sample (*n* = 196) *n* (%)
**Sex**	
Male	24 (12.2%)
Female	172 (87.8%)
**Age**	
18–30 years	55 (28.1%)
31–41 years	33 (16.8%)
41–65 years	108 (55.1%)
**Work experience**	
1–10 years	80 (40.8%)
11–20 years	25 (12.8%)
21–30 years	56 (28.6%)
31–40 years	35 (17.9%)
**Manager or employee?**	
Manager	8 (4.1%)
Employee	188 (95.9%)
**Educational level**	
Elementary school	3 (1.5%)
Highschool graduation	16 (8.2%)
Qualified nurse	113 (57.7%)
Nurse BSc.	48 (24.5%)
Nurse MSc.	16 (8.2%)
**Income**	
Under 200,000 HUF	6 (3.1%)
200–300,000 HUF	42 (21.4%)
300–400,000 HUF	76 (38.8%)
400–500,000 HUF	56 (28.6%)
Above 500,000 HUF	16 (8.2%)
**Current university nursing studies?**	
Yes	43 (21.9%)
No	153 (78.1%)

**Table 2 healthcare-12-01857-t002:** Distribution of data related to the COVID-19 pandemic (*n* = 196).

	Total Sample (*n* = 196) *n* (%)
**How many hours did your work on COVID ward per a month?**	
20–120 h	18 (9.2%)
121–180 h	114 (58.2%)
above 180 h	64 (32.7%)
**How many hours did you work on night shift in COVID ward per a month?**	
I did not work on night shift	18 (9.2%)
12–24 h	8 (4.1%)
24–48 h	12 (6.1%)
48–72 h	54 (27.6%)
72–120 h	73 (37.2%)
120 h or more	31 (15.8%)
**How many waves of the coronavirus pandemic did you work through?**	
1.	17 (8.7%)
2.	35 (17.9%)
3.	52 (26.5%)
4.	41 (20.9%)
5.	51 (26%)

**Table 3 healthcare-12-01857-t003:** Correlation between the probability of staying in the healthcare career within 5 years ranges on a scale of 1 to 10, and certain questions of the Motivation at Work Scale. questionnaire. (1: definitely not staying; 10: definitely staying) (*n* = 196).

Because I enjoy this work very much	Spearman Correlation	0.304
	*p* value	0.000
	*n*	196
Because I have fun doing my job	Pearson Correlation	0.257
	*p* value	0.000
	*n*	196
For the moments of pleasure that this job brings me	Pearson Correlation	0.222
	*p* value	0.002
	*n*	196
I chose this job because it allows me to reach my life goals	Pearson Correlation	0.261
	*p* value	0.000
	*n*	196
Because this job fulfils my career plans	Pearson Correlation	0.329
	*p* value	0.000
	*n*	196
Because this job fits my personal values	Pearson Correlation	0.279
	*p* value	0.000
	*n*	196
Because I have to be the best in my job, I have to be a “winner”	Pearson Correlation	0.155
	*p* value	0.030
	*n*	196
Because my work is my life, and I don’t want to fail	Pearson Correlation	0.197
	*p* value	0.006
	*n*	196
Because my reputation depends on it	Pearson Correlation	0.111
	*p* value	0.120
	*n*	196
Because this job affords me a certain standard of living	Pearson Correlation	0.225
	*p* value	0.002
	*n*	196
Because it allows me to make a lot of money	Pearson Correlation	0.195
	*p* value	0.006
	*n*	196
I do this job for the paycheck	Pearson Correlation	0.017
	*p* value	0.809
	*n*	196

**Table 4 healthcare-12-01857-t004:** Outcomes of the multiple linear regression analysis, which predicts staying another 5 years in the profession (*n* = 196).

Coefficients ^a^						
Model		Unstandardized Coefficients		Standardized Coefficients	t	Sig.
		B	Std. Error	Beta		
1	(Constant)	7.068	1.319		5.358	0.000
	Intinsic motivation	−0.066	0.104	−0.061	−0.629	0.530
	Identified motivation	0.427	0.105	0.427	4.073	0.000
	Introjected motivaton	0.066	0.072	0.068	0.916	0.361
	External regulation	0.032	0.076	0.030	0.424	0.672
	Monthly income	0.169	0.193	0.066	0.877	0.382
	Work experience	0.090	0.032	0.441	2.839	0.005
	Age	−0.088	0.034	−0.388	−2.574	0.011
	Dummy_Current_Study_Yes	0.437	0.423	0.075	1.034	0.302
	Dummy_Sufficient_Staffing_No	−0.572	0.466	−0.085	−1.227	0.222
	Dummy_Sufficient_Protective equipment	0.567	0.364	0.111	1.559	0.121
	Total score of fear of reassignment to a COVID ward questionnaire	−0.199	0.134	−0.102	−1.484	0.140

^a^ Dependent Variable: Please rate the likelihood that you will continue to work in healthcare for at least 5 years on a scale from 1 to 10.

## Data Availability

The data presented in this study are available upon request from the corresponding author.

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
