# Peer review of "Work Motivation: A Wall That Not Even the COVID-19 Pandemic Could Knock Down: Research Article"

_healthcare, 2024, doi:10.3390/healthcare12181857_

Round 1
Reviewer 1 Report
Comments and Suggestions for Authors
Dear authors,
I have read your paper. The theme is quite interesting and you investigated 196 persons using developed questionnaires.
Although I see some interesting points, I really believe that this paper needs to be developed to a greater extent in order to be published.
First, we need strong theoretical background around motivation and other employee attitudes. You beed to explain theoretically these issues and to add some results from previous similar research on this topic.
Second, please explain Sample before results. Two long paragraphs are related to the sample description and this is not a result but just description of your sample.
Third, the biggest challenge is the data processing approach. Only desriptive sratustics and correlations are used. I suggest to try with more comprehensive techniques like regression models or even SEM.
Fourth, you need to prove the validity of the data and how you deal with the cmb.
Fifth, discussion and conclusion need to be rewritten after you perform new statistics.
Comments on the Quality of English Language
N/A
Reviewer 2 Report
Comments and Suggestions for Authors
Thank you for the opportunity to review your paper. I hope you find these suggestions helpful.
Abstract
· Good description of the study.
· Line 35 – should be working in “a” or “the” COVID ward.
· Line 36 – should be Data and “anonym” should be anonymous (if it truly was) or confidential if it wasn’t fully anonymous.
· Line 37 grammar
Background Lit Review
Good background, contextual, and theoretical information.
Methods
Additional information regarding methods is recommended as follows:
· Sample. How were participants recruited? How was informed consent provided. Define health care provider.
· Measures. Please describe your measures, their properties, and data collection tools.
· Data Collection. Place measures information above. Describe how the participants linked to the survey. How many questions total? How long did it take to complete the survey? How many did you invite but declined/did not complete? Assume you meant SPSS “25” not “2.5.”
· Somewhere, more description of the study context is needed.
Results
· Overall, the results are text heavy and repeat what is in table/figures. I would suggest seeing if these can be cleaned up/streamlined.
· In describing the sample, consider a table.
· Table 2 should include SDs. Should not be a table but rather titled “Figure”
· Table 3 title has a typo “,,” and should note this is an average score? If yes, should include SDs. Should not be a table but rather titled “Figure”
· Table 4 – same as above.
· Table 5 is squished – some text is deleted.
Discussion
· Throughout the discussion, citations are written incorrectly: For example - sake of the goal. (Ryan & Deci, 2000) – it should be: sake of the goal (Ryan & Deci, 2000). Please fix throughout manuscript.
· The discussion may repeat results more than what is needed. I would suggest reviewing this section for deletion of redundancies.
References
· Several extra spaces to be fixed in copy editing.
Comments on the Quality of English LanguageNeeds significant copy editing and APA reference/citation editing.
Reviewer 3 Report
Comments and Suggestions for Authors
This study focus on the the motivating factors among nurses serving during the coronavirus pandemic when most studies were discussing the factors influencing career abandonment, and identified a significant correlation between intrinsic motivation and the probability of leaving the profession within 5 years. We think that this topic actually is not so innovative but it still has certain research value. I have just some minor comments.
1. Abstract and Introduction: The authors needed to give the full name of the COVID-19 and other english abbreviations at its first appearance in abstract and introduction (main text). And second, statistical symbols such as n, r, and p, should be italicized in our study.
2. Introduction: In the first paragraph, the author spent a lot of ink describing the number of registered healthcare workers in European countries, but in reality, the focus should be on the turnover rate.
3. Introduction: In “1.2. Motivation”, if possible, the author should increase the research status of work motivation, not necessarily in the context of COVID-19.
4. Materials and Methods: The authors were advised to give the Cronbach’s alpha coefficient of the original “Motivation at Work Scale” questionnaire and the Cronbach’s alpha coefficient in the present study. Besides, in the “Data analysis” section, t-test and ANOVA are mentioned, please elaborate further on whether the data conform to a normal or approximately normal distribution.
5. Results: For the sociodemographic information of study participants, authors were recommended to only list the important information and create a table for the remaining information, rather than presenting it all in text. Besides, please pay attention to symbols with same meaning should remain consistent throughout the text, such as N and n. In addition, should Table 1~4 be modified to Figure 1~4?
6. Results: In “3.1. Result of the Motivation at Work Scale”, the results section only presents objective results, but should not include interpretation of the findings which should be added to the discussion section. I will not emphasize the same issue in the following text, and the author should pay attention to making revisions.
7. Table 5: “The probability of staying in the healthcare career within 5 years ranges on a scale of 1 to 10 (1: Definitely not staying in the career / 10: Definitely staying in)”, authors were advised to place this sentence in the table note instead of vertically placing it in the table.
8. Discussion: In general, research hypotheses should be presented in the introduction or methodology section, rather than in the discussion section.
9. Discussion: In 4.1, 4.2, 4.3, and 4.5, the discussion needs to go beyond just describing study findings again and stating consistency with previous research and expanding on possible causes, etc., and overall, it is not deep enough.
10. Maybe the author can provide some intervention suggestions based on the research findings to help managers motivate nurses to stay in their profession and some study suggestions for future related research.
11. Overall, I believe the manuscript may benefit from a thorough review of the language.
Comments on the Quality of English LanguageMinor editing of English language required.
Round 2
Reviewer 1 Report
Comments and Suggestions for Authors
Please, add section about sample before results. Presentation of the sample is not a result. Put it in section 2.5 Sample and make 2.6. data analysis.
Author Response
Dear Reviewer!
I have made the requested changes! Please find the attached revised manuscript! I hope this time it will meet your expectations!
Reviewer 3 Report
Comments and Suggestions for Authors
No objections, agree to publish.
Author Response
Dear Reviewer!
Thank you very much for your comment!